# Elevated Hedgehog-Interacting Protein Levels in Subjects with Prediabetes and Type 2 Diabetes

**DOI:** 10.3390/jcm8101635

**Published:** 2019-10-06

**Authors:** An-Chi Lin, Hao-Chang Hung, Yun-Wen Chen, Kai-Pi Cheng, Chung-Hao Li, Ching-Han Lin, Chih-Jen Chang, Hung-Tsung Wu, Horng-Yih Ou

**Affiliations:** 1Division of Endocrinology and Metabolism, Department of Internal Medicine, National Cheng Kung University Hospital, College of Medicine, National Cheng Kung University, Tainan 70403, Taiwanhaochang.hung@gmail.com (H.-C.H.); supercabyhome@yahoo.com.tw (K.-P.C.); cyclops0113@yahoo.com.tw (C.-H.L.); 2Department of Pharmacology, College of Medicine, National Cheng Kung University, Tainan 70403, Taiwan; yunwen_chen@mail.ncku.edu.tw; 3Department of Health Management Center, National Cheng Kung University Hospital, National Cheng Kung University, Tainan 70403, Taiwan; smallhear@gmail.com; 4Department of Family Medicine, National Cheng Kung University Hospital, College of Medicine, National Cheng Kung University, Tainan 70403, Taiwan; changcj.ncku@gmail.com; 5Graduate Institute of Metabolism and Obesity Sciences, College of Nutrition, Taipei Medical University, Taipei 11031, Taiwan

**Keywords:** hedgehog-interacting protein, impaired fasting glucose, impaired glucose tolerance, newly diagnosed diabetes, normal glucose tolerance

## Abstract

Background: The prevalence of diabetes is rapidly increasing worldwide and is highly associated with the incidence of cancers. In order to prevent diabetes, early diagnosis of prediabetes is important. However, biomarkers for prediabetes diagnosis are still scarce. The hedgehog-interacting protein (Hhip) is important in embryogenesis and is known to be a biomarker of several cancers. However, Hhip levels in subjects with diabetes are still unknown. Methods: In total, 314 participants were enrolled and divided into normal glucose tolerance (NGT; *n* = 75), impaired fasting glucose (IFG; *n* = 66), impaired glucose tolerance (IGT; *n* = 86), and newly diagnosed diabetes (NDD; *n* = 87) groups. Plasma Hhip levels were determined by an ELISA. The association between the Hhip and the presence of diabetes was examined by a multivariate linear regression analysis. Results: There were significant differences in the body mass index, systolic and diastolic blood pressure, fasting plasma glucose (FPG), post-load 2-h glucose, hemoglobin A1c (A1C), C-reactive protein, total cholesterol, triglyceride, and high- and low-density lipoprotein cholesterol levels among the groups. Concentrations of the Hhip were 2.45 ± 2.12, 4.40 ± 3.22, 4.44 ± 3.64, and 6.31 ± 5.35 ng/mL in subjects in the NGT, IFG, IGT, and NDD groups, respectively. In addition, we found that A1C and FPG were independently associated with Hhip concentrations. Using NGT as a reference group, IFG, IGT, and NDD were all independently associated with Hhip concentrations. Conclusions: Hhip was positively associated with prediabetes and type 2 diabetes mellitus.

## 1. Introduction

According to statistics from the International Diabetes Federation, the global diabetes prevalence was 8.8% in 2017 and is expected to further increase to 9.9% by 2045 [1]. Since type 2 diabetes increases the risks of liver and pancreatic cancers, as well as the risks of bladder, breast, and colorectal cancers by 20%–50% [2]. Diabetes has become an important public health issue. In addition, people with diabetes have a higher risk of developing cardiovascular diseases, renal failure, retinopathies, and neuropathies. Because of the increased economic and financial burden of diabetes and related complications, strategies for controlling and preventing diabetes are important issues for investigation.

Prediabetes is a state of abnormal glucose homeostasis characterized by the presence of impaired fasting glucose (IFG) and impaired glucose tolerance (IGT) [3]. Individuals with prediabetes are at increased risks for type 2 diabetes and cardiovascular diseases compared to individuals with normal glucose tolerance (NGT) [4]. Elevated plasma glucose that characterizes IGT should be measured 2 h after a 75 g oral glucose tolerance test (OGTT) given in the morning after an overnight fast. OGTT may be better at diagnosing postprandial pathology than hemoglobin A1c (A1C) during the early stages. However, it requires patient preparation prior to the test, which causes clinical inconvenience. From a clinical perspective, the most serious drawback of the OGTT is its lack of reproducibility. This is why both the American Diabetes Association (ADA) and the World Health Organization require a second OGTT to confirm a diagnosis of diabetes [5]. Thus, the investigation of biomarkers is a novel strategy for diagnosing prediabetes.

The hedgehog pathway is a key signaling pathway for the maintenance and regeneration of various tissues in embryogenesis. The hedgehog-interacting protein (Hhip) is a 700-residue transmembrane glycoprotein that functions as a negative regulator in the hedgehog pathway [6]. It has been well-demonstrated that renal Hhip expression is associated with nephropathy development in diabetes and that hyperglycemia-induced renal Hhip expression may mediate glomerular endothelial fibrosis and apoptosis in diabetes [7]. In addition, the Hhip is associated with several cancers including gastrointestinal cancer, hepatocellular carcinoma, and pancreatic cancer [8,9,10,11,12,13]. Diabetes is highly associated with incidences of cancers and the Hhip is also associated with cancer development. However, the role of Hhip in the pathogenesis of diabetes remains unclear.

Previous studies demonstrated that loss of Hhip expression results in significant impairment of pancreatic morphogenesis, islet formation, and endocrine cell proliferation. A genome-wide diabetes profiling database revealed that compared to lean animals, Hhip messenger (m)RNA seems to be significantly upregulated in islets of diabetic ob/ob mice [14]. In addition, the Hhip inhibits insulin secretion in mice by altering islet integrity in response to high-fat diet-mediated beta cell dysfunction [15]. As beta cell failure is a major cause of type 2 diabetes, we hypothesized that the Hhip might be associated with glycemia. Thus, we tested our hypothesis by measuring soluble Hhip protein levels in subjects with different glycemic statuses.

## 2. Methods

### 2.1. Study Participants

All participants in this case-control study were screened between January and December 2016 at National Cheng Kung University Hospital (Tainan, Taiwan). Participants signed an informed consent form approved by the Institutional Review Board of National Cheng Kung University Hospital (ER-104-204).

After a 12 h overnight fast, all subjects received a blood test, including routine biochemistry. All healthy subjects who did not have a medical history of diabetes received a standard 75 g OGTT. To avoid the confounding effects of age and sex, we selected subjects with NGT, IFG, IGT, and newly diagnosed diabetes (NDD) in the order of their admission to the study. NGT, IFG, IGT, and NDD were defined according to ADA criteria: NGT, as fasting plasma glucose (FPG) of <5.6 mmol/L and 2 h post-load glucose of <7.8 mmol/L without a history of diabetes; IFG, as FPG of 5.6–7.0 mmol/L and 2 h post-load glucose of <7.8 mmol/L; IGT, as FPG of <5.6 mmol/L and 2 h post-load glucose of 7.8–11.0 mmol/L; and diabetes, as FPG of ≥7.0 mmol/L or 2 h post-load glucose of ≥11.1 mmol/L. Among the study cohort, 314 participants were enrolled and classified into the NGT (*n* = 75), IFG (*n* = 66), IGT (*n* = 86), and NDD (*n* = 87) groups.

Subjects with the following conditions or diseases were excluded: (1) Any acute or chronic inflammatory disease as determined by a leukocyte count of >10,000/mm^3^ or clinical signs of infection; (2) any other major diseases, including generalized inflammation or advanced malignant diseases contraindicating this study; and (3) pregnancy.

Each subject’s body height (to the nearest 0.1 cm), weight (BW; to the nearest 0.1 kg), and waist circumference (WC; to the nearest 0.1 cm) were measured. The body mass index (BMI) was calculated as the weight (in kilograms) divided by the height (in meters) squared. For blood pressure measurements, subjects were asked to rest in a supine position in a quiet ambience, and measurements were obtained in a fasting state between 08:00 and 10:00. Two blood pressure readings, separated by an interval of at least 5 min, were taken with an appropriate-sized cuff wrapped around the right upper arm using a DINAMAP vital signs monitor (model 1846SX; Critikon, Irvine, CA, USA). Blood glucose was measured by the hexokinase method (Roche Diagnostic, Mannheim, Germany). Serum insulin (Mercodia AB, Uppsala, Sweden) was measured by an enzyme-linked immunosorbent assay (ELISA). High-sensitivity C-reactive protein (hsCRP) was measured using a highly sensitive ELISA kit (Immunology Consultants Laboratory, Newberg, OR, USA). The determination of Plasma Hhip was carried out using a Human Hhip ELISA kit (MyBioSource, San Diego, CA, USA). Serum total cholesterol (TC), triglyceride (TG), and high-density lipoprotein cholesterol (HDL-C) levels were determined in the central laboratory of National Cheng Kung University Hospital with an autoanalyzer (Hitachi 747E; Tokyo, Japan). A1C was measured with a high-performance liquid chromatographic method (Tosoh Automated Glycohemoglobin Analyzer; Tokyo, Japan).

### 2.2. Statistics

SPSS software (version 17.0; SPSS, Chicago, IL, USA) was used for all statistical analyses. Clinical characteristics are expressed as the mean ± standard deviation (SD) or as a percentage. Chi-squared and *t*-tests were used to analyze the difference in categorical and continuous variables among groups, respectively. The Kruskal-Wallis test was used for comparison of the Hhip and TG levels among groups. A multivariate linear regression analysis was conducted to identify independent predictors of the Hhip concentration. Independent variables included age, gender, TC, TGs, HDL-C, creatinine, hsCRP, NDD, hypertension, and the BMI. A *p* value of <0.05 was considered statistically significant.

## 3. Results

In total, 314 subjects, 75 with NGT, 66 with IFG, 86 with IGT, and 87 with NDD, were enrolled. Comparisons of basic characteristics of study subjects are shown in Table 1. There were significant differences in the BMI (*p* = 0.049), systolic blood pressure (*p* = 0.003), diastolic blood pressure (*p* = 0.025), FPG (*p* < 0.001), post-load 2 h glucose (*p* < 0.001), A1C (*p* < 0.001), hsCRP (*p* = 0.001), HDL-C (*p* = 0.006), TG (*p* = 0.006), LDL-C (*p* = 0.021), and TC (*p* = 0.038) levels among the four groups. Concentrations of the Hhip were 2.45 ± 2.12, 4.40 ± 3.22, 4.44 ± 3.64, and 6.31 ± 5.35 ng/mL in NGT, IFG, IGT, and NND subjects, respectively. In the post-hoc analysis, plasma Hhip concentrations were significantly higher in the IFG (*p* = 0.018), IGT (*p* = 0.008), and NDD (*p* < 0.001) groups compared to the NGT group (Figure 1). In addition, there were significant differences in plasma Hhip concentrations between the IFG and NDD groups (*p* = 0.016) and between the IGT and NDD groups (*p* = 0.009). However, there was no significant difference in plasma Hhip concentrations between the IFG and IGT groups.

To further determine independent factors associated with plasma Hhip concentrations, we performed a multivariate linear regression analysis (Table 2). Using NGT as the reference group, we found that A1C (β = 0.175, 95% CI = 0.207–0.894, *p* = 0.002) was independently associated with Hhip concentrations after adjusting for confounding factors listed in model 1 of Table 1. When substituting A1C for FPG in model 2, we found that FPG (β = 0.012, 95% CI = 0.001–0.024, *p* = 0.032) was independently associated with Hhip concentrations. Finally, to investigate the independent effect of the glycemic status on circulating Hhip concentrations, we used NGT as the reference group and found that IFG (β = 0.236, 95% CI = 1.086–3.864, *p* = 0.001), IGT (β = 0.214, 95% CI = 0.690–3.103, *p* = 0.002), and NDD (β = 0.404, 95% CI = 2.305–4.830, *p* < 0.001) were all independently associated with Hhip concentrations (model 3). In contrast, insulin (β = −0.143, 95% CI = (−0.0397)–(−0.045), *p* = 0.014) was negatively associated with Hhip concentrations (model 3).

## 4. Discussion

To the best of our knowledge, this is the first study to investigate correlations between plasma Hhip concentrations and glycemic groups. We found that the Hhip was positively associated with prediabetes, including IFG and IGT, as well as type 2 diabetes.

Previous studies demonstrated that elevated levels of hedgehog signaling block pancreas formation [16,17,18]. Landsman et al. also demonstrated that in mouse studies, increased hedgehog signaling led to impairment of beta cell differentiation [19]. The Hhip functions as an inhibitor of hedgehog signaling within developing organs of the fore-midgut region, including the pancreas [18]. Both type 1 diabetes (T1D) and T2D are characterized by a deficit in the pancreatic beta cell mass [20]. Insulin secretion was markedly reduced in pancreatic diabetic patients with the lowest beta cell area [21]. Chronic hyperglycemia leads to beta cell hypertrophy and loss of beta cell differentiation [22,23]. Thus, we supposed that in a hyperglycemic status, Hhip expression is increased as compensation to promote differentiation of beta cells. Nchienzia et al. recently demonstrated that the pancreatic Hhip gene inhibits insulin secretion by altering islet integrity in beta cells [15]. This is compatible with our finding that insulin was negatively associated with Hhip concentrations.

In addition to the pancreas, the Hhip also plays an important role in the development of insulin-sensitive tissues, such as muscles and adipocytes. Ochi et al. demonstrated that the Hhip regulates muscle development by sequestering hedgehog [24]. The Hhip also inhibits proliferation and promotes differentiation of adipocytes through suppressing the hedgehog signaling pathway [25]. These findings suggest that the Hhip might play an important role in energy utilization regulated by both skeletal muscles and adipose tissues.

There are some limitations in this work. First, the cross-sectional design of this study did not allow for causal inference between plasma Hhip concentrations and the development of diabetes. Second, we could not directly measure changes in human pancreatic tissue cells to evaluate beta cell differentiation. Third, the measurement of Hhip did not run in duplicate. Fourth, some borderline significantly statistical results, such as BMI of a *p*-value of 0.045, may be at risk of type 1 error. Fifth, hedgehog signaling dysfunction is also involved in such as nonalcoholic fatty liver disease, asthma, and chronic obstructive pulmonary disease. The specificity of this biomarker in glycemia might be decrease. Finally, all the study subjects were ethnic Chinese (i.e., Taiwanese) and the findings might not be generalizable to other ethnicities.

## 5. Conclusions

The Hhip was positively associated with prediabetes and type 2 diabetes mellitus, and the evaluation of Hhip as a potential biomarker for progression of glycemia need further study.

## Figures and Tables

**Figure 1 jcm-08-01635-f001:**
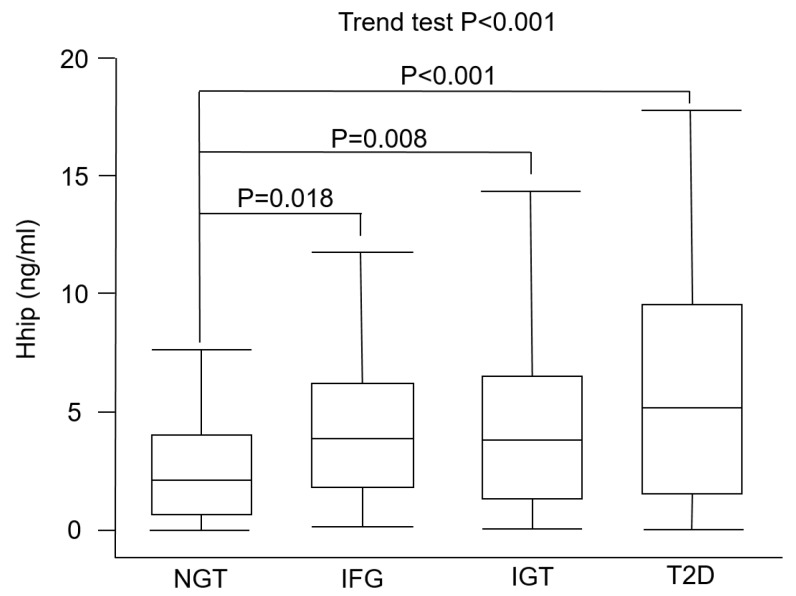
Plasma concentrations of the hedgehog-interacting protein (Hhip) in subjects with prediabetes or diabetes. Box and whisker plot of plasma Hhip concentrations in participants with normal glucose tolerance (NGT; *n* = 75), impaired fasting glucose (IFG; *n* = 66), impaired glucose tolerance (IGT; *n* = 86), and type 2 diabetes (T2D; *n* = 87). The line inside the box represents the median of the distribution, the box top and bottom values are defined by the 25th and 75th percentiles, and the whiskers are minimum and maximum values.

**Table 1 jcm-08-01635-t001:** Comparisons of clinical parameters among subjects with normal glucose tolerance (NGT), impaired fasting glucose (IFG), impaired glucose tolerance (IGT), and newly diagnosed diabetes (NDD).

	NGT	IFG	IGT	NDD	*p* Value
*N*	75	66	86	87	
Age (years)	61.5 ± 12.4	62.4 ± 12.0	62.4 ± 11.9	62.7 ± 11.6	NS
Male (%)	50.7	65.2	55.8	57.5	NS
Hhip (ng/mL)	2.45 ± 2.12	4.40 ± 3.22	4.44 ± 3.64	6.31 ± 5.35	<0.001
BMI (kg/m^2^)	22.3 ± 2.7	23.6 ± 2.8	23.4 ± 3.0	23.3 ± 3.2	0.049
SBP (mmHg)	121.4 ± 17.2	126.9 ± 17.2	128.7 ± 18.2	131.9 ± 19.8	0.003
DBP (mmHg)	70.4 ± 9.9	73.4 ± 10.2	73.7 ± 10.7	75.4 ± 10.9	0.025
FPG (mmol/L)	85.8 ± 7.3	104.2 ± 4.8	88.5 ± 11.6	138.4 ± 61.6	<0.001
Post-load 2-h glucose (mmol/L)	97.0 ± 24.2	107.0 ± 22.7	161.7 ± 16.7	262.1 ± 87.5	<0.001
A1C (%)	5.7 ± 0.3	5.8 ± 0.4	5.8 ± 0.3	7.3 ± 2.0	<0.001
AST (U/L)	26.6 ± 8.0	27.1 ± 14.0	24.7 ± 7.9	30.7 ± 43.6	NS
ALT (U/L)	23.5 ± 10.5	27.5 ± 21.1	22.5 ± 10.7	32.7 ± 54.9	NS
hsCRP (mg/L)	1.9 ± 3.0	3.0 ± 5.5	3.0 ± 5.3	5.9 ± 10.0	0.001
Total cholesterol (mmol/L)	197.1 ± 36.8	205.8 ± 31.6	196.2 ± 36.3	211.4 ± 48.7	0.038
Triglyceride (mmol/L) *	93.9 ± 38.1	119.0 ± 67.5	113.4 ± 60.2	131.3 ± 81.8	0.006
HDL-C (mmol/L)	60.5 ± 19.2	53.5 ± 15.1	53.5 ± 14.5	52.6 ± 13.8	0.006
LDL-C (mmol/L)	117.8 ± 35.7	128.5 ± 29.7	120.0 ± 31.9	132.5 ± 39.5	0.021

Data are expressed as the mean ± standard deviation (SD) or as a percentage. * Values were log-transformed before analysis. Hhip, hedgehog-interacting protein; BMI, body-mass index; SBP, systolic blood pressure; DBP, diastolic blood pressure; A1C, hemoglobin A1c; AST, aspartate aminotransferase; ALT, alanine aminotransferase; FPG, fasting plasma glucose; hsCRP, high-sensitivity C-reactive protein; HDL-C, high-density lipoprotein cholesterol; LDL-C, low-density lipoprotein cholesterol.

**Table 2 jcm-08-01635-t002:** Results of a multivariate linear regression analysis between Hhip and clinical variables.

Variable	Model 1		Model 2		Model 3	
	β (95% CI)	*p*	β (95% CI)	*p*	β (95% CI)	*p*
Age	−0.015 (−0.043, 0.032)	NS	−0.027 (−0.047, 0.028)	NS	−0.085 (−0.072, 0.015)	NS
Sex	0.171 (0.513, 2.298)	0.002	0.173 (0.519, 2.321)	0.002	0.124 (0.059, 1.938)	0.037
A1C (%)	0.175 (0.207, 0.894)	0.002				
FPG (mmol/L)			0.120 (0.001, 0.024)	0.032		
IFG vs. NGT					0.236 (1.086, 3.864)	0.001
IGT vs. NGT					0.214 (0.690, 3.103)	0.002
NDD vs. NGT					0.404 (2.305, 4.830)	<0.001
Insulin (mIU/L)					−0.143 (−0.397, 0.045)	0.014
hsCRP (mg/L)					0.067 (−0.027, 0.106)	NS
BMI (kg/m^2^)					0.003 (−0.160, 0.169)	NS
SBP (mmHg)					−0.004 (−0.027, 0.025)	NS
eGFR					0.010 (−0.023, 0.027)	NS
ALT (U/L)					−0.064 (−0.022, 0.006)	NS
Total cholesterol (mmol/L)					−0.027 (−0.096, 0.091)	NS
* Triacylglycerol (mmol/L)					−0.048 (−6.922, 5.091)	NS
HDL-C (mmol/L)					−0.070 (−0.117, 0.103)	NS
LDL-C (mmol/L)					0.076 (−0.086, 0.103)	NS

A1C, hemoglobin A1c; IFG, impaired fasting glucose; NGT, normal glucose tolerance:, NDD, newly diagnosed diabetes; SBP, systolic blood pressure; DBP, diastolic blood pressure; ALT, alanine aminotransferase; FPG, fasting plasma glucose; hsCRP, high-sensitivity C-reactive protein; BMI, body-mass index; eGFR, estimated glomerular filtration rate; HDL-C, high-density lipoprotein cholesterol; LDL-C, low-density lipoprotein cholesterol. * Values were log-transformed before analysis.

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
