# Peer review of "Elevated Hedgehog-Interacting Protein Levels in Subjects with Prediabetes and Type 2 Diabetes"

_jcm, 2019, doi:10.3390/jcm8101635_

Round 1

Reviewer 1 Report

This is an interesting paper about between serum Hhip concentrations and glycemic groups.  

The authors found that the Hhip was positively associated with prediabetes, including IFG and IGT, as well as type 2 diabetes.

Overall, this manuscript is well-written. The methods and analysis are appropriate and results accurately presented.

The hedgehog pathway dysregulation is also involved in pathogenesis of other common prevalent and chronic diseases such as non-alcoholic liver disease, chronic obstructive pulmonary disease, and asthma.  Given its high prevalence, the specificity of this biomarker could decrease. Readers would appreciate a brief comment about it.

Author Response

Response to Reviewer 1 Comments

This is an interesting paper about between serum Hhip concentrations and glycemic groups. 

 The authors found that the Hhip was positively associated with prediabetes, including IFG and IGT, as well as type 2 diabetes.

 Overall, this manuscript is well-written. The methods and analysis are appropriate and results accurately presented.

 The hedgehog pathway dysregulation is also involved in pathogenesis of other common prevalent and chronic diseases such as non-alcoholic liver disease, chronic obstructive pulmonary disease, and asthma.  Given its high prevalence, the specificity of this biomarker could decrease. Readers would appreciate a brief comment about it.

 Response: Thank you for the valuable comment. We have adding this information as a limitation (Page 7, line 208-210)

Reviewer 2 Report

The authors present an analysis of serum Hhip protein levels in a set of patients ranging from normal to new onset diabetes. A few points to consider:

abstract

-you state logistic regression but your methods say linear with the later being correct. Please correct abstract

-are your hhip levels with s.d.?

Body of manuscript:

-your hypothesis that hhip may be a suitable biomarker wasn't really tested in your research. Yes, it was associated with differing degrees of glucose tolerance/impairement but you do not report the sensitivity or specificity as a potential biomarker or discuss its utility compared to currently accepted diagonstic markers. Consider clarifying your objective as only looking for associations with hhip and list evaluations of biomarker potential as a future directoin; or add certain elements that supports hhip as a biomarkers and why it has more utility than current biomarkers (e.g., glucose, insulin, a1c, etc.).

-how did you ensure that subjects had newly diagnosed diabetes? please explain in methods

-what was the company for the insulin elisa's?

-were elisa runs performed in replicate. If so, list how many, if not, acknowledge as a potential limitatoin

-A limitation that should be addressed is the considerable number of statistical tests you have without a correction. I am not necessarily saying you have to perform bonferroni but you should acknowledge when some borderline ( such as bmi and p=0.045 or potentially some of the hhip comparisons, may be at risk of type 1 error.

-Again, in the results specifically state if you are listing S.D.'s or S.E's in text and table 1. 

-Consider listing specific p-values for hhip comparisons since this is your main comparisons in the study

-were hhip normally distributed?

-How were the models chosen/constructed in the multivariate section?

-As stated above, there should be expanded discussion on the potential utility of hhip as an early pre-diabetic marker including limitatoins

Author Response

Response to Reviewer 2 Comments

The authors present an analysis of serum Hhip protein levels in a set of patients ranging from normal to new onset diabetes. A few points to consider:

 abstract

 -you state logistic regression but your methods say linear with the later being correct. Please correct abstract

Response : Thank you. We have corrected the abstract as “linear” regression.

 -are your hhip levels with s.d.?

Response : Thank you. The hhip levels are presented with mean ± standard deviation (SD). We have provided the information in the STATISTICS section (Page 3, line 130).

 Body of manuscript:

 -your hypothesis that hhip may be a suitable biomarker wasn't really tested in your research. Yes, it was associated with differing degrees of glucose tolerance/impairement but you do not report the sensitivity or specificity as a potential biomarker or discuss its utility compared to currently accepted diagonstic markers. Consider clarifying your objective as only looking for associations with hhip and list evaluations of biomarker potential as a future directoin; or add certain elements that supports hhip as a biomarkers and why it has more utility than current biomarkers (e.g., glucose, insulin, a1c, etc.).

Response : Thank you for the valuable comment. We have clarifying our objective as only to looking for associations between Hhip and glycemia (Page 2, line 86-87), and list evaluations of biomarker potential as a future direction (Page 6, line 209).

 -how did you ensure that subjects had newly diagnosed diabetes? please explain in methods

Response : Thank you for the comment. All healthy subjects who did not have a medical history of diabetes received a 75g oral glucose tolerance test. Glycemic status were defined according to ADA criteria: normal glucose tolerance (NGT), as fasting plasma glucose  of <5.6 mmol/l and 2-h post-load glucose of <7.8 mmol/l without a history of diabetes; impaired fasting glucose (IFG), as FPG of 5.6~7.0 mmol/l and 2-h post-load glucose of <7.8 mmol/l; impaired glucose tolerance (IGT), as FPG of <5.6 mmol/l and 2-h post-load glucose of 7.8~11.0 mmol/l; and newly diagnosed diabetes (NDD), as FPG of ≥7.0 mmol/l or 2-h post-load glucose of ≥11.1 mmol/l. We have described this information in the METHOD section (Page 3, line 98-104).

 -what was the company for the insulin elisa's?

Response : Thank you for the comment. The ELISA kits were purchased from Mercodia AB, Uppsala, Sweden. This information had been described in the METHOD section (Page 3, line 119).

 -were elisa runs performed in replicate. If so, list how many, if not, acknowledge as a potential limitatoin

Response : Thank you for the comment. The ELISA did not run in duplicate, and we have added this as a limitation (Page 6, line 204).

 -A limitation that should be addressed is the considerable number of statistical tests you have without a correction. I am not necessarily saying you have to perform bonferroni but you should acknowledge when some borderline ( such as bmi and p=0.045 or potentially some of the hhip comparisons, may be at risk of type 1 error.

Response : Thank you for the comment. We have adding the risk of type 1 error in the limitation (Page 7, line 207).

 -Again, in the results specifically state if you are listing S.D.'s or S.E's in text and table 1.

Response : Thank you for the comment. All the results in the study were represented as mean ± standard deviation (SD) or as a percentage, and this information have been expressed in the STATISTICS section (Page 3, line 130), and Table 1 (Page 4, line 152).

 -Consider listing specific p-values for hhip comparisons since this is your main comparisons in the study

Response : Thank you for the comment. We have specific the p-values for Hhip comparisons (Page 4, line 145-147; Figure 1)

 -were hhip normally distributed?

Response : Thank you for the comment. Since the p-value of less than 0.001 was noted in Shapiro-Wilk test, the Hhip levels were not normally distributed. Therefore, we used Kruskal-Wallis test for comparison of Hhip among different glycemic groups.

 -How were the models chosen/constructed in the multivariate section?

Response : Thank you for the comment. The variables used in Model 1 and Model 2 were age, sex, and glycemia (A1C and FPG, respectively), which were chosen based on our hypothesis that Hhip might be associated with glycemia. The variables used in Model 3 were  clinical variables that have been associated with glycemia in previous studies.

 -As stated above, there should be expanded discussion on the potential utility of hhip as an early pre-diabetic marker including limitatoins

Response : Thank you for the valuable comment. Indeed, our study did not address the utility of Hhip as a biomarker. We have clarifying our objective as only to looking for associations between Hhip and glycemia (Page 2, line 86-87), and list evaluations of biomarker potential as a future direction (Page 6, line 209).